# MixSize: Training Convnets With Mixed Image Sizes for Improved Accuracy, Speed and Scale Resiliency

## Abstract

Convolutional neural networks (CNNs) are commonly trained using a fixed spatial image size predetermined for a given model. Although trained on images of a specific size, it is well established that CNNs can be used to evaluate a wide range of image sizes at test time, by adjusting the size of intermediate feature maps.

In this work, we describe and evaluate a novel mixed-size training regime that uses several image sizes at training time. We demonstrate that models trained using our method are more resilient to image size changes and generalize well even on small images. This allows faster inference by using smaller images at test time. For instance, we receive a $76.43\%$ top-1 accuracy (ResNet50 on ImageNet) with an image size of 160, which matches the accuracy of the baseline model with $2\times$ fewer computations. Furthermore, for a target image size used at test time, we show this method can be exploited either to accelerate training or improve the final test accuracy. For example, we are able to reach a $79.27\%$ accuracy with the same model evaluated at a 288 spatial size for a relative improvement of $14\%$ over the baseline. MixSize regimes pave the way for faster and more accurate training and inference using convolutional networks. Our PyTorch implementation and pre-trained models are publicly available[1].

## 1 Introduction

Convolutional neural networks are successfully used to solve various tasks across multiple domains such as visual (Krizhevsky et al., 2012; Ren et al., 2015), audio (van den Oord et al., 2016), language (Gehring et al., 2017) and speech (Abdel-Hamid et al., 2014). While scale-invariance is considered important for visual representations (Lowe, 1999), convolutional networks are not scale invariant with respect to the spatial resolution of the image input, as a change in image dimension may lead to a non-linear change of their output. Even though CNNs are able to achieve state-of-the-art results in many tasks and domains, their sensitivity to the image size is an inherent deficiency that limits practical use cases, and requires that images at evaluation time match training image size. For example, Touvron et al. (2019) demonstrated that networks trained on specific image size, perform poorly on other image sizes at evaluation time, as confirmed in Figure 1.

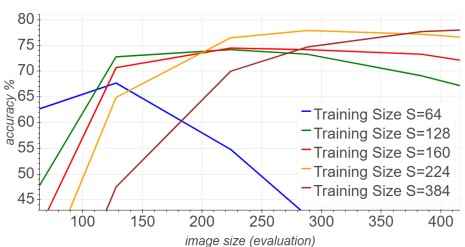

Figure 1: Top-1 test accuracy per image size, models trained on specific sizes (ResNet50, ImageNet).

The most common method to improve scale invariance in CNNs to artificially enlarge the dataset using a set of label-preserving transformations also known as "data augmentation" (Howard, 2013; Krizhevsky et al., 2012). Several of these transformations scale and crop objects appearing within the data, thus increasing the network's robustness to inputs of different scale. Several works attempted to achieve scale invariance by modifying the network structure to learn over multiple possible target input scales (Takahashi et al., 2017; Xu et al., 2014; Zhang et al., 2019). These methods

---

[1] https://github.com/paper-submissions/mixsize

explicitly change the model for specific input size, thus not benefiting from possible lower computational requirements of using smaller image sizes, nor with ability of inferring on sizes not observed during training. Another approach suggested by Cai et al. (2020) modifies network structure and training regime to account for variety of inference modes without additional specialized training.

In this work, we introduce a novel training regime, "MixSize" for convolutional networks that uses stochastic image and batch sizes. The main contributions of the MixSize regime are:

- **Reducing image size sensitivity**. We show that the MixSize training regime can improve model performance on a wide range of sizes used at evaluation.

- **Faster inference**. As our mixed-size models can be evaluated at smaller image sizes, we show up to $2\times$ reduction in computations required at inference to reach the same accuracy as the baseline model.

- **Faster training vs. high accuracy**. We show that reducing the average image size at training leads to a trade-off between the time required to train the model and its final accuracy.

## 2 BACKGROUND AND RELATED WORK

### 2.1 USING MULTIPLE IMAGE SIZES

Deep convolutional networks are traditionally trained using fixed-size inputs, with spatial dimensions $H \times W$ and a batch size $B$. The network architecture is configured such that the spatial dimensions are reduced through strided pooling or convolutions, with the last classification layer applied on a $1 \times 1$ spatial dimension. Modern convolutional networks usually conclude with a final "global" average pooling (Lin et al., 2013; Szegedy et al., 2015), which reduces any remaining spatial dimensions with a simple averaging operation. Modifying the spatial size of an input to a convolutional layer by a factor $\gamma$, will yield an output with size scaled by the same factor $\gamma$. This modification does not require any change to the number of parameters of the given convolutional layer, nor its underlying operation. It was observed by practitioners and previous works that a network trained on a specific input dimension can still be used at inference using a modified image size to some extent (Simonyan & Zisserman, 2014). Moreover, evaluating with an image size that is larger than used for training can improve accuracy up to a threshold, after which it quickly deteriorates (Touvron et al., 2019).

Although not explicitly trained to handle varying image sizes, CNNs are commonly evaluated on multiple scales post-training, such as in the case of detection (Lin et al., 2017; Redmon & Farhadi, 2018; Liu et al., 2020) and segmentation (He et al., 2017) tasks. In these tasks, a network that was pretrained with fixed image size for classification is used as the backbone of a larger model that is expected to adapt to a wide variety of image sizes.

Recently, Tan & Le (2019) showed a computation-vs-accuracy trade-off in applying scaling to the image size used for convolutional networks training and evaluation. This finding is consistent with past findings, which demonstrated that training with a larger image size can result in a better classification accuracy (Huang et al., 2018; Szegedy et al., 2016). In addition, previous works explored the notion of "progressive resizing" (Howard, 2018; Karras et al., 2017) — increasing image size as training progresses to improve model performance and time to convergence. A similar idea by Wu et al. (2020) was used to improve performance of training on video data, by balancing resolution with batch size. Another related work by Touvron et al. (2019) demonstrated that CNNs can be trained using a fixed small image size and fine-tuned post-training to a larger size, with which evaluation will be performed. This procedure reduced the train-test discrepancy caused by the change in image size and allowed faster training time and improved accuracy — at the cost of additional fine-tuning procedure and additional computations at inference time.

In this work we will further explore the notion of using multiple image sizes at training, so the CNN performance will be resilient to test time changes of the image size.

### 2.2 LARGE BATCH TRAINING OF DEEP NETWORKS

Deep neural network training can be distributed across many computational units and devices. The most common distribution method is by "data-parallelism" — computing an average estimate of the

gradients using multiple, separately computed data samples. As training NN models is done using batch-SGD method and its variants, scaling this process across multiple computational devices while maintaining similar utilization for each device inflates the global batch size.

Large batch training is known to affect the generalization capabilities of NNs and as such, it requires to modify its optimization regime. While several works claimed that large-batch training leads to an inherent "generalization gap" (Keskar et al., 2016), more recent works demonstrated that this gap is largely caused by insufficient number of optimization steps performed and can be partly mitigated by hyper-parameter tuning (Hoffer et al., 2017; Shallue et al., 2018). In order to cope with the variations in the network training dynamics, several modifications of the optimization procedure have been proposed. For instance, a linear (Goyal et al., 2017) or square-root (Hoffer et al., 2017) scaling of the learning rate with respect to the batch size growth. Other modifications include per-layer gradient scaling schemes (You et al., 2017) and optimizer modifications (Ginsburg et al., 2019). Several works also explored using incremented batch-sizes (Smith et al., 2018) in order to decrease the number of training iterations required to reach the desired accuracy.

Recent work (Hoffer et al., 2020) introduced the notion of "Batch Augmentation" (BA) — increasing the batch size by augmenting several instances of each sample within the same batch. BA aids generalization across a wide variety of models and tasks, with the expense of an increased computational effort per step. A similar method called "Repeated Augmentation" (RA) was proposed by Berman et al. (2019). It was also demonstrated that BA may allow to decrease the number of training steps needed to achieve a similar accuracy and also mitigate I/O throughput bottlenecks (Choi et al., 2019). As previous works investigated mostly homogeneous training settings (e.g., using a fixed batch size), an open question still exists on the utility of rapidly varying batch-sizes. We will explore this notion by modifying the optimization method, enabling training with varying batch-sizes and limited hyper-parameter tuning.

## 3 MixSize: Training with multiple image scales

The traditional practice of training convolutional networks using fixed-size images holds several shortcomings. As it is a common practice to use different image size for evaluation than that used for training (He et al., 2017; Lin et al., 2017; Redmon & Farhadi, 2018), it was observed by Touvron et al. (2019) and empirically verified (Figure 1) that classification accuracy may degrade above or below a certain size threshold. Similarly, we observed this phenomenon in a wide variety of common networks (Appendix Figure 4). This hints the issue here is related more to the training procedure, rather than to a specific model. To remedy these issues, we suggest a stochastic training regime, where image sizes may vary in each optimization step.

**Motivation.** We hypothesize that a considerable part of the training of convolutional networks can be performed at a lower image size than the target size used at test time. As a first step to substantiate this claim, we evaluated the impact of various image sizes on the CNN training progress by examining gradient statistics during training. Specifically, we measured the correlation between weight gradients with respect to varying image sizes (Appendix Table 2). We see that gradients computed across different sizes of the same images have a stronger correlation than those obtained across different images. Moreover, the correlation is most significant during the first stages of training and decreases as the model converges. This observation suggests that smaller image gradients can be used as an estimate to the full image gradients, with a smaller computational footprint. Therefore, using large images along the entire training process may be sub-optimal in terms of computational resource utilization. We continue to test our hypothesis using a novel mixed-size training regime.

**MixSize training regime.** We propose "MixSize", a stochastic training regime where input sizes can vary. In this regime, we modified the spatial dimensions $H, W$ (height and width) of the input image[2], as well as the batch size. The batch size is modified either by the number of used samples, denoted $B$, or by the number of batch-augmentations for each sample (Hoffer et al., 2020), denoted $D$ ("duplicates"). To simplify our notation and use-cases, we will follow the common practice of training on square images and use $S = H = W$. Formally, in the MixSize regime, these

---

[2]The spatial dimensions of all intermediate maps in the CNN are changed accordingly, at the same scale as the input.

sizes can be described as random variables sharing a single discrete distribution

$$(\hat{S}, \hat{B}, \hat{D}) = \{(S, B, D)_i \ \ w.p. \ \ p_i\} \,, \tag{1}$$

where $\forall i : p_i \geq 0$ and $\sum_i p_i = 1$.

As the computational cost of each training step is approximately proportional to $S^2 \cdot B \cdot D$, we choose these quantities to reflect a nearly fixed budget for any choice $i$ such that $S_i^2 B_i D_i \approx Const$. Thus, in this regime, the computational and memory requirements for each step in remain fixed.

**Benefits and Trade-offs.** We will show that using our MixSize regime can have a positive impact on the resiliency of trained networks to various images sizes at evaluation time by demonstrating better accuracy across a wide range of sizes. This entails a considerable saving in computational burden required for inference, especially when using smaller models. Furthermore, given a fixed budget of computational and time resources (per step), we can now modify our regime along spatial and batch axes. We will explore two trade-offs:

- **Decrease number of iterations per epoch** – by enlarging $B$ at the expense of $S$.
- **Improve generalization per epoch** – by enlarging $D$ at the expense of $S$.

We denote our modified mixed-size regimes as $B^+$ for an increased effective batch-size and $D^+$ for an increased number of BA duplicates of the same ratio.

## 4    IMPROVED TRAINING PRACTICES FOR MIXSIZE

MixSize regimes continuously change the statistics of the model's inputs, by modifying the image size as well as batch-size. This behavior may require hyper-parameter tuning and may also affect size-dependent layers such as batch normalization (Ioffe & Szegedy, 2015). To easily adapt training regimes to the use of MixSize as well as improve their final performance, we continue to describe three methods we found useful: **step-wise size sampling**, **gradient smoothing** and **batch-norm calibration**.

### 4.1    SAMPLING IMAGE SIZE PER-STEP

Considering our findings on correlation of gradients across different image size in early training, we were initially motivated to consider a training variation where size is progressively enlarged (Howard, 2018; Karras et al., 2017).
In this method we consider increasing image size from small to large, where each image size is used for a predefined number of epochs out of total number of training epochs required.
We observed progressive resizing to cause noticeable accuracy deterioration when switching one size for another (Appendix E), which may be attributed to the network tendency to adapt to a specific size. Unfortunately, this hampered use cases where the final image size was larger than the one used at evaluation.

We therefore considered two alternatives for image size variations, where size is randomly sampled and changed throughout training:

- Sampling image size per epoch
- Sampling image size per training step

These sampling methods can be seen as a stochastic alternative to long and short cycle multigrid approach suggested by Wu et al. (2020). We found that random sampling regimes perform better than scaling image size from small to large, keeping accuracy high across all used sizes at evaluation (Appendix E). While sampling both at epoch and step time frames performed similarly, replacing sizes on each step converged faster and exhibited less noise in measured test accuracy. Considering these findings, we continue to perform our experiments using the third regime – sampling image size per training step.

## 4.2 Gradient smoothing

Training with varying batch and spatial sizes inadvertently leads to a change in the variance of the accumulated gradients. For example, in Table 2, the gradient variance is larger when computed over a small image size (unsurprisingly). This further suggests that the optimization regime should be adapted to smaller spatial sizes, in a manner similar to learning-rate adaptations that are used for large-batch training. This property was explored in previous works concerning large-batch regimes, in which a learning rate modification was suggested to compensate for the variance reduction for larger batch-sizes. Unfortunately, the nature of this modification can vary from task to task or across models (Shallue et al., 2018), with solutions such as a square-root scaling (Hoffer et al., 2017), linear scaling (Goyal et al., 2017) or a fixed norm ratio (You et al., 2017). Here we suggest changing both the spatial size as well as the batch size, which is also expected to modify the variance of gradients within each step and further complicates the choice of optimal scaling.

Previous works suggested methods to control the gradient norm by gradient normalization (Hazan et al., 2015) and gradient clipping (Pascanu et al., 2013). These methods explicitly disable or limit the gradient's norm used for each optimization step, but also limit naturally occurring variations in gradient statistics. We suggest an alternative solution to previous approaches, which we refer to as "Gradient smoothing". Gradient smoothing mitigates the variability of gradient statistics when image sizes are constantly changing across training.

Denoting $L$ for the objective loss function, $g_t$ and $w_t$ as weights and gradients of optimization step $t$, we introduce an exponentially moving weighted average of the gradients' norm $\bar{g}_t$ (scalar) which is updated according to

$$\bar{g}_t = \alpha\bar{g}_{t-1} + (1 - \alpha)g_t$$

where

$$g_t = \left\|\frac{\partial L}{\partial w_t}\right\|_2 \quad \text{and} \quad \bar{g}_0 = g_0\,.$$

We normalize the gradients used for each step by the smoothing coefficient, such that each consecutive step is performed with gradients of similar norm. For example, for the vanilla SGD step, we use a weight update rule of the form

$$w_{t+1} = w_t - \eta\frac{\bar{g}_t}{g_t}\frac{\partial L}{\partial w_t}\,.$$

This running estimate of gradient norm is similar to the optimizer suggested by Ginsburg et al. (2019), which keeps a per-layer estimate of gradient moments. Gradient smoothing, however, is designed to adapt globally (across all layers) to the batch and spatial size modification and can be used regardless of the optimization method used.

We found gradient smoothing to be beneficial in $B^+$ regimes, where multiple varying batch sizes are used, as the effective learning rate is also modified according to common practices. We observed that gradient smoothing reduces the gap between gradient norms of different sizes (see Appendix Figure 6) and allow the use of an average learning rate scaling (weighted across batch sizes). Measuring test error on the same model shows a significant advantage for gradient-smoothing (Appendix Figure 7).

## 4.3 Batch-norm calibration for varying image sizes

As demonstrated by Touvron et al. (2019), using a different image size at evaluation may incur a discrepancy between training and evaluation protocols, caused by using different data pre-processing. (Touvron et al., 2019) suggested a post-training procedure, where a network trained on a specific fixed-size is fine-tuned on another size, later used for evaluation. Their solution required dozens of training epochs, amounting to thousands of full forward and back-propagation computations, along with parameter updates for batch-norm and classifier layers. In contrast, we surmise that for networks trained with mixed-regimes, discrepancy issues mainly arise from the use of the batch-norm layers (Ioffe & Szegedy, 2015) and can be solved by targeting them specifically.

Batch-norm layers introduce a discrepancy between training and test evaluations (Ioffe, 2017), as at inference a running estimate of the mean and variance (of training data) are used instead of the actual mean and variance values. This difference is emphasized further in the use of varying image

size, as changing the spatial size of an input map can significantly modify the measured variance of that map. While a fine-tuning process per image size can eliminate this discrepancy (Touvron et al., 2019), it required significant amount of computation and labeled data. We offer a simpler alternative. For each evaluated size, we calibrate the mean and variance estimates used for that size by computing an average value over a small number of training examples. This calibration requires only a few (hundreds) of feed-forward operations with no back-propagation or parameter update, and takes only a few seconds on a single GPU. As calibration requires very little resources, as well as small amount of unlabelled data, it can be potentially performed on-demand at test-time.

Interestingly, we highlight the fact that although this process has little to no effect on models trained using a fixed-size input, it does improve our mixed-size models considerably on a wide range of image sizes.

## 5 EXPERIMENTS

### 5.1 MIXSIZE WITH A FIXED IMAGE SIZE AT TEST-TIME: THE SPEED-ACCURACY TRADE-OFF

We used our sampling strategy to train and compare our regime to the baseline results of various models over the CIFAR10/100 (Krizhevsky, 2009) and ImageNet (Deng et al., 2009) visual classification tasks. In each experiment we use the original hyper-parameters without modification. For the $B^+$ regime, use our gradient smoothing method, as described in Section 4.2.

As CIFAR datasets are limited in size, we consider the following balanced stochastic regime chosen:

$$S^{(28)} : \ S = \begin{cases} 40, & \text{w.p.} \quad p = 0.2 \\ 32, & \text{w.p.} \quad p = 0.3 \\ 24, & \text{w.p.} \quad p = 0.3 \\ 16, & \text{w.p.} \quad p = 0.2 \end{cases}$$

The regime was designed to be centered around the mean value of 28. As the original image size used for training is $32 \times 32$, we are now able to increase either the batch size or number of BA duplicates for each training step by a factor of $\frac{32^2}{S^2}$ such that $S^2 \cdot B \cdot D$ is approximately constant.

For the ImageNet dataset, we use the following stochastic regime found by cross-validation on several alternatives over the ResNet-50 model (He et al., 2016):

$$S^{(144)} : \ S = \begin{cases} 256, & \text{w.p} \quad p = 0.1 \\ 224, & \text{w.p} \quad p = 0.1 \\ 128, & \text{w.p} \quad p = 0.6 \\ 96, & \text{w.p} \quad p = 0.2 \end{cases}$$

While the original training regime consisted of images of size $224 \times 224$, our proposed regime makes for an average image size of $\bar{S} \times \bar{S} = 144 \times 144$. This regime was again designed so that the reduced spatial size can be used to increase the corresponding batch size or the number of BA duplicates, as described in Section 3.

For each result, we measure our final test accuracy on the original image size ($32 \times 32$ for CIFAR, $224 \times 224$ for ImageNet). We also perform batch-norm calibration as described in Section 4.3.

We present the final test accuracy, as well as the number of training steps for each model in Table 1. We see that our MixSize regimes yield two possible improvements:

- Reduced number of training steps to achieve a similar test accuracy using $B^+$ regime.
- Better test accuracy using $D^+$ regime (enabling BA with same computational budget).

We note that for both $B+$ and $D+$ regimes, the final accuracy matches or surpasses the baseline results. $B+$ enables larger batches on average, hence the reduction in training steps, while $D+$ leverages benefits of batch-augmentation to achieve better accuracy within the same compute. The ResNet-50 model benefits the most from our MixSize regime on the ImageNet model and so we chose it for further exploration.

Table 1: Test accuracy (Top-1) results for CIFAR and ImageNet. Each row represents models trained using the same computational and memory budget per step. Steps and accuracy are reported at the completion of a fixed epoch budget (i.e., 90 epochs for ResNet on ImageNet, 200 for ResNet on CIFAR). Accuracy is reported for model's original size (32 for CIFAR, 224 for ImageNet).

| Network | Dataset | Steps | | | Accuracy | | |
|---|---|---|---|---|---|---|---|
| | | Baseline | $B^+$ | $D^+$ | Baseline | $B^+$ | $D^+$ |
| ResNet-44 | CIFAR10 | 156K | **109K** | 156K | 92.84% | 94.30% | **94.46%** |
| WRN-28-10 | CIFAR10 | 156K | **109K** | 156K | 96.60% | 97.28% | **97.68%** |
| AmoebaNet | CIFAR10 | 469K | **328K** | 469K | 98.16% | 98.16% | **98.32%** |
| ResNet-44 | CIFAR100 | 156K | **109K** | 156K | 70.36% | 72.19% | **73.10%** |
| WRN-28-10 | CIFAR100 | 156K | **109K** | 156K | 79.85% | 83.08% | **83.52%** |
| ResNet-50 | ImageNet | 450K | **169K** | 450K | 76.40% | 76.87% | **78.04%** |
| DenseNet169 | ImageNet | 450K | **169K** | 450K | 76.20% | 76.66% | **77.26%** |
| EfficientNet-B0 | ImageNet | 1000K | **376K** | 1000K | 76.32% | 76.37% | **76.53%** |

## 5.2 Case study: ResNet-50 in mixed size regimes

We showcase our finding on the common ResNet-50 model (He et al., 2016) using the training regime suggested by Goyal et al. (2017) (see Appendix C.2). We employed the standard data augmentation and *did not incorporate any additional regularization or augmentation techniques.*

We are first interested in accelerating the time needed for convergence of the tested models using our $B^+$ scheme. We enlarge the batch size used for each spatial size by a factor of $\frac{224^2}{S^2}$ such that $S^2 \cdot B$ is kept approximately fixed. As the average batch size is larger than $B_o$, which was used with the original optimization hyper-parameters, we scale the learning rate linearly as suggested by Goyal et al. (2017) by a factor of $\frac{\bar{B}}{B_o}$. We note that for the proposed regimes we did not require any learning rate warm-up, due to the use of gradient smoothing.

As can be seen in Figure 2, regime $B^+$ enables training with approximately $2.7\times$ fewer training steps, while reaching an improved accuracy of 76.61%. As sizes were chosen to reflect in approximately equal computational cost per iteration, $B^+$ regime offers a similar improvement in total wall-clock time.

Next, we perform a similar experiment with a $D^+$ regime, where the number of BA duplicates is similarly increased with respect to $D_o$ instead of the batch size. This scaling results with an average duplicates of $\bar{D} = 3$.

As the computational cost for each step remains approximately constant, as well as the number of required steps per epochs, training a model under this regime requires an equal wall-clock time. However,

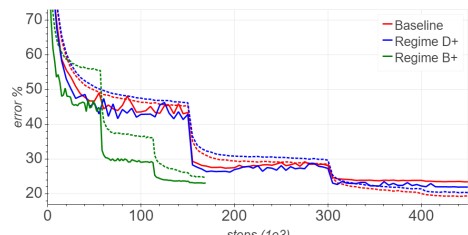

Figure 2: Training (dotted) and test accuracy on ImageNet using suggested regimes (224 evaluation size). All regimes required similar resources per step. $B^+$ regime required $\approx 2.7\times$ less steps per epoch.

the increased batch-augmentation improves the final test accuracy to 78.04%, approximately 7% relative improvement over the 76.4% baseline.

## 5.3 Increasing model resiliency to test-time changes in image size

Next, we examine how MixSize affects the resulting model resiliency to changes in the image size during test-time. We evaluated the models by varying the test-time image sizes around the original 224 spatial size: $S = 224 + 32 \cdot m$, for $m \in \{-6, ..., 6\}$. The common evaluation procedure for ImageNet models first scales the image to a 256 smallest dimension and crops a center $224 \times 224$ image. We adapt this regime for other image sizes by scaling the smallest dimension to $\lfloor \frac{8}{7} S \rfloor$ (since $\frac{8}{7} \cdot 224 = 256$) and then cropping the center $S \times S$ patch. Models trained with a mixed regime were calibrated to a specific evaluation size by measuring batch-norm statistics for 200 batches of

training samples. We note that for original fixed-size regimes this calibration procedure resulted with degraded results and so we report accuracy without calibration for these models. We did not use any fine-tuning procedure post training for any of the models.

As can be seen in Figure 3a, the baseline model trained using a fixed size, reaches $76.4\%$ top-1 accuracy at the same $224$ spatial size it was trained on. As observed previously, the model continues to slightly improve beyond that size, to a maximum of $76.8\%$ accuracy. However, it is apparent that the model's performance quickly degrades when evaluating with sizes smaller than $224$.

We compare these results with $D^+$ regimes, trained with an average size of $\bar{S} = 144$ and $\bar{S} = 208$ (see Appendix F). As described earlier, this model requires the same time and computational resources as the baseline model. However, due to the decreased average size, we were able to leverage more than 1 duplicates per batch on average, which improved the model's top-1 accuracy to $77.14\%$ at size $224$. Furthermore, we find that the model performs much more favorably at sizes smaller than $224$, scoring an improved (over baseline) accuracy of $76.43\%$ at only $160$ spatial size.

The model trained with the $S^{(208)}$ regime offers a similar improvement in accuracy, but across a larger spatial size, as it observed an average size of $208$ during training. Figure 3a demonstrates that while all three models (Fixed with $S = 224$, $S^{(144)}$ and $S^{(208)}$) were trained with the same compute and memory budget, mixed-size regimes offer superior accuracy over a wide range of evaluation sizes. Specifically, mixed-regime at $S = 208$ dominates the baseline fixed-size regime at all sizes, while our mixed regime at $S = 144$ achieves best results at sizes smaller than $224$.

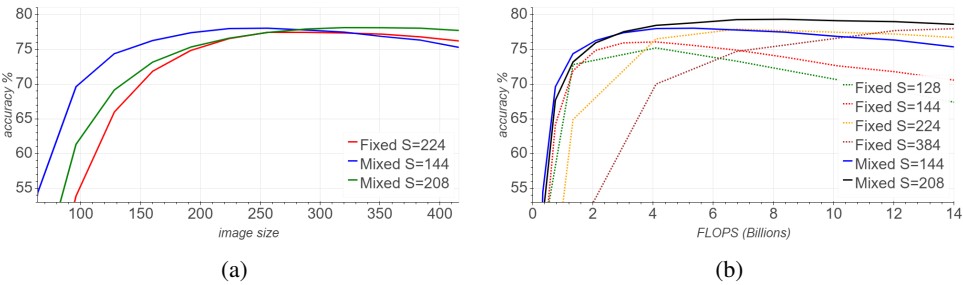

|     |     |
| :-: | :-: |
| (a) | (b) |

Figure 3: (a): Test accuracy on validation set per image size, all models trained using the same computational and memory resources (regime $D^+$). (b): Test accuracy per Gflop (at evaluation).

We also compared the classification performance across evaluated image sizes, using networks trained on a variety of fixed sizes and our mixed regimes. As a baseline, we use results obtained by Touvron et al. (2019) (trained with RA, without fine-tuning) and compare them with mixed-regime models trained with an equal computational budget, by setting the base number of BA duplicates to $D = 2$. As can be seen in Figure 3b, mixed-regime trained models offer a wider range of resolutions with close-to-baseline accuracy (within a $2\%$ change) and perform better than their fixed-size counterparts at all sizes. As the number of floating-point operations (flops) grows linearly with the number of pixels, using a mixed regime significantly improves accuracy per compute at evaluation. We further note that our $S^{(224)}$ model reaches a top accuracy of $79.27\%$ at a $288$ evaluation size.

## 6  SUMMARY

In this work, we introduced and examined a performance trade-off between computational load and classification accuracy governed by the input's spatial size. We suggested stochastic image size regimes, which randomly change the spatial dimension as well as the batch size and the number of augmentation duplicates in the batch. Stochastic regime benefits are threefold: (1) reduced number of training iterations; or (2) improved model accuracy (generalization) and (3) improved model robustness to changing the image size. We believe this approach may have a profound impact on the practice of training convolutional networks. Given a computational and time budget, stochastic size regimes may enable to train networks faster, with better results, as well as to target specific image sizes that will be used at test time. As the average size chosen to train is reflected in the optimal operating point for evaluation resolution, mixed regimes can be used to create networks with better performance across multiple designated use cases.

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

# Appendix

## A  SCALE RESILIENCY IN PRETRAINED CONVOLUTIONAL NETWORKS

We used 6 common networks (Simonyan & Zisserman, 2014; Szegedy et al., 2015; He et al., 2016; Huang et al., 2017; Mahajan et al., 2018; Sandler et al., 2018) pretrained on the ImageNet dataset at a resolution of $224 \times 224$ and measured their accuracy on varying sizes. As can be seen, these networks show similar patters of sensitivity to the evaluation size, suggesting this is caused by the training protocol (which is similar for all), rather than the used model.

## B  GRADIENT CORRELATION OVER IMAGE SIZES

We evaluated the impact of various image sizes on the CNN training progress by examining gradient statistics during training (Table 2). We used a ResNet-44 model (He et al., 2016), trained on the CIFAR10 dataset (Krizhevsky, 2009), whose standard image size is $32 \times 32$. Measurements are performed on the whole network's gradient vector. Images were sampled in uniform. The smaller $24 \times 24$ images were down-sampled with bilinear interpolation.

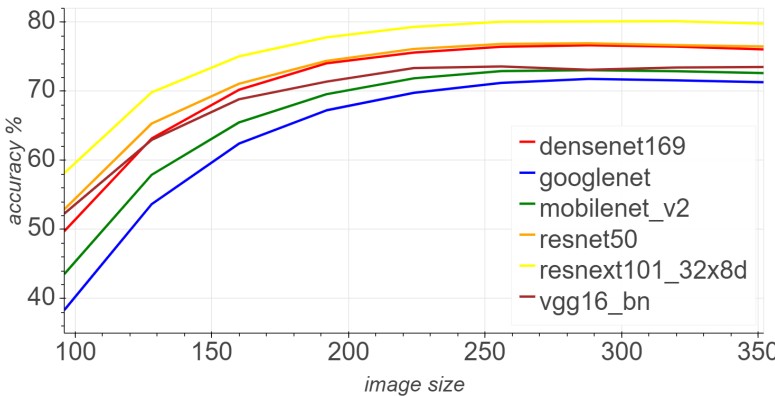

Figure 4: Top-1 accuracy over ImageNet validation set per evaluation image size.

Table 2: ResNet-44 gradient correlation on CIFAR10. We measure the Spearman correlation coefficient $\rho$ between gradients computed over different spatial size of random images $\rho\left(g_x^{(s_1)}, g_x^{(s_2)}\right)$, as well as non-identical random images of the same size $\rho\left(g_x^{(s_1)}, g_y^{(s_1)}\right)$. We also compute the variance $V(g_x)$ for the gradients of each spatial size.

| Measure | Network State | | |
|---|---|---|---|
| | Initial | Partially Trained | Fully Trained |
| Epoch | 1 | 50 | 100 |
| Test Accuracy | 55.12% | 87.56% | 92.62% |
| $\rho\left(g_x^{(32)}, g_x^{(24)}\right)$ | 0.2 | 0.08 | 0.03 |
| $\rho\left(g_x^{(32)}, g_y^{(32)}\right)$ | 0.086 | 0.02 | $-0.004$ |
| $V\left(g_x^{(32)}\right)$ | $1.03e^{-6}$ | $1.44e^{-6}$ | $6.24e^{-7}$ |
| $V\left(g_x^{(24)}\right)$ | $1.95e^{-6}$ | $6.34e^{-6}$ | $2.26e^{-5}$ |

## C EXPERIMENTAL SETTINGS

### C.1 CIFAR

We examine our method using the common visual datasets CIFAR10/100 (Krizhevsky, 2009) that consist of $32 \times 32$ color images. We use the ResNet-44 model suggested by (He et al., 2016), Wide Resnet WRN-28-10 (Zagoruyko, 2016) and AmoebaNet (Real et al., 2019) with their original regime and batch size of $64$. While for ResNet-44 we use the original augmentation protocol as described by He et al. (2016). In this method, the input image is padded with $4$ zero-valued pixels at each side, top and bottom. A random $32 \times 32$ part of the padded image is then cropped and with a $0.5$ probability flipped horizontally. In order to adapt to varying input scales, we add an additional augmentation step, that resizes the images using bilinear interpolation to $S \times S$, depending on a sampled size for each step. We note that this keeps the exact original augmentation procedure for $S = 32$. For WRN-28-10 and AmoebaNet we apply cutout (DeVries & Taylor, 2017) and auto-augment policies (Cubuk et al., 2018) .

Training progress on the CIFAR10 using ResNet44 is depicted in Figure 5. Interestingly, although designed only to reduce training time, we can see that our $B^+$ regime also improves accuracy in some cases. This improvement can be attributed to a regularization effect induced by changing image sizes during training, also manifested by an increase in training error throughout its progress.

### C.2 IMAGENET

We used the ResNet-50 model (He et al., 2016) with the training regime suggested by (Goyal et al., 2017) that consists of base learning rate of $0.1$, decreased by a factor of $10$ on epochs $30, 60, 80$, stopping at epoch $90$. We used the base batch size of $256$ over $4$ devices and $L_2$ regularization

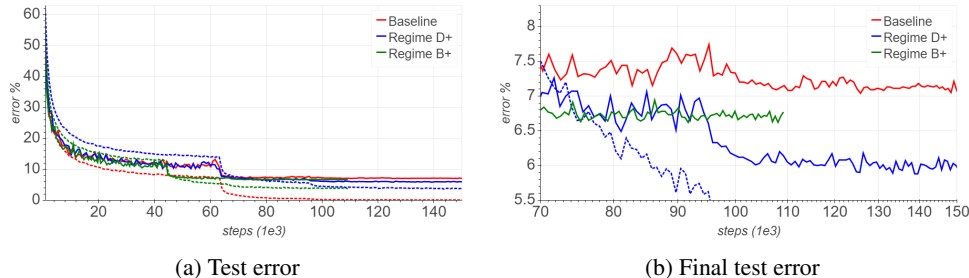

(a) Test error  (b) Final test error

Figure 5: Training (dotted) and test accuracy vs optimization step (ResNet44, CIFAR10). We compare vanilla training with two computationally equivalent stochastic regimes: increased duplicates ($D^+$) and increased batch ($B^+$). $B^+$ **regime achieves better test accuracy at a reduced number of iterations, while** $D^+$ **improves accuracy further at a similar computational cost**.

over weights of convolutional layers. We employed the standard data augmentation and did not incorporate any additional regularization or augmentation techniques.

Additionally, we also used the EfficientNet-B0 model suggested by (Tan & Le, 2019). We used the same data augmentation and regularization as the original paper, but opted for a shorter training regime with a momentum-SGD optimizer that consisted of a cosine-annealed learning rate (Loshchilov & Hutter, 2016) over 200 epochs starting from an initial base $0.1$ value.

## D    IMPACT OF GRADIENT SMOOTHING

We show the impact of gradient smoothing on $B^+$ MixSize regimes on CIFAR10 6 and ImageNet 7. We can see that gradient smoothing helps to reduce gap between gradient norms at difference batch sizes and improves training and validation accuracy.

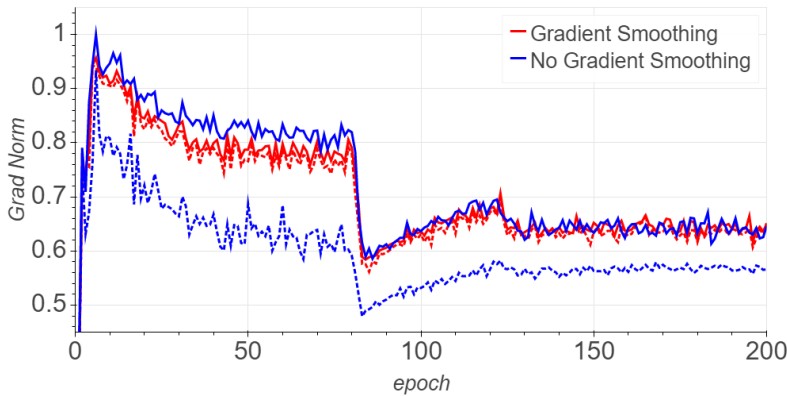

Figure 6: Impact of gradient smoothing on CIFAR10, ResNet-44. The training regime includes two image sizes: $32 \times 32$ and $16 \times 16$ (average size is $S = 24$). Using a $B^+$ regime creates two batch sizes: $256$ and $2,048$ respectively. Gradient norm values are shown with and without Grad-smoothing. Solid lines are gradients for $S = 32$ while dotted lines are for $S = 16$. Gradient smoothing helps to reduce gap between gradient norms at difference batch sizes.

## E    COMPARING MIXSIZE REGIMES

We consider training regimes with varying image sizes, such that the average image size is smaller than the desired evaluation size. For example, for the height dimension $H$, we wish to obtain an average size of $\bar{H} = \sum_i p_i H_i$ such that $\bar{H} < H_o$. We consider three alternatives for image size variations:

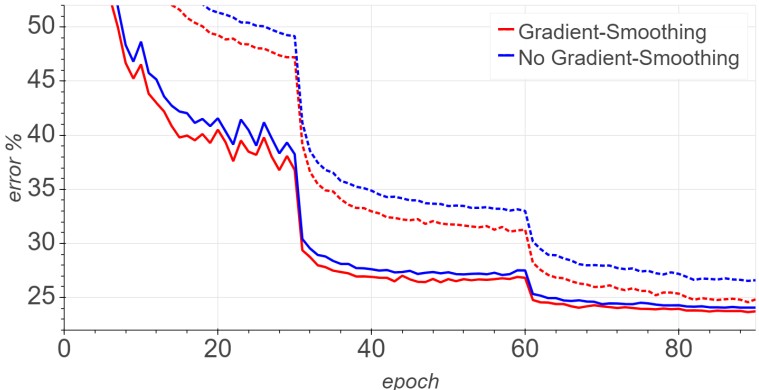

Figure 7: Impact of gradient smoothing on ImageNet, ResNet50. Shown are the validation and training (dashed) top-1 accuracy using gradient smoothing and without it. The training regime used is $B^+$ using an average size of $144 \times 144$ ($s^{(144)}$

- Increase image size from small to large (progressive resizing), where each image size is used for number of epochs $E_i = p_i E_{\text{total}}$, where $E_{\text{total}}$ is the total number training epochs required.

- Using a random image size for each epoch, keeping the epoch number for each size at $E_i$

- Sampling image size per training step at probability $p_i$

We used a $B^+$ regime and adapt learning rate for each batch-size according to best practices (Hoffer et al., 2017; Goyal et al., 2017). In both CIFAR10 (Figure 8) and ImageNet (Figure 9) sampling methods exhibit better accuracy than a progressive resizing regime. This is most apparent when the final size used at training is larger than target size for inference (on which test accuracy is measured).

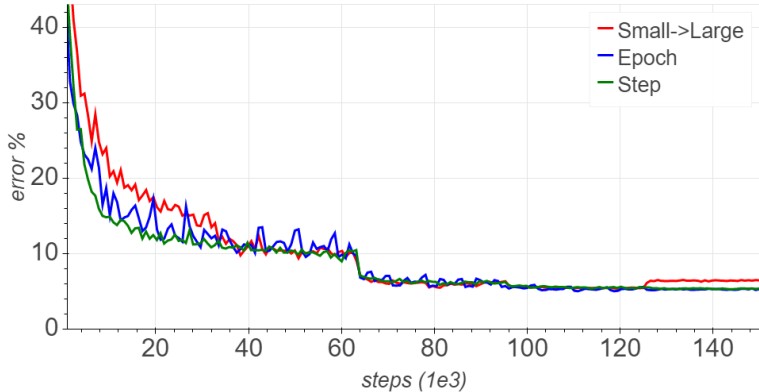

Figure 8: Test accuracy vs step (ResNet44 on CIFAR10, $B^+$ with $s^{(28)}$) for 3 size sampling regimes: (1) From small to large (2) Sample each Epoch (3) Sample each step. All methods reached a similar accuracy, but sampling each epoch was less noisy and did not require hyper-parameter tuning.

## F  ALTERNATIVE SIZE DISTRIBUTION REGIMES

We used alternative size regimes balanced around 224, named $S^{(208)}$ and $S^{(224)}$. They can be described by the following distributions:

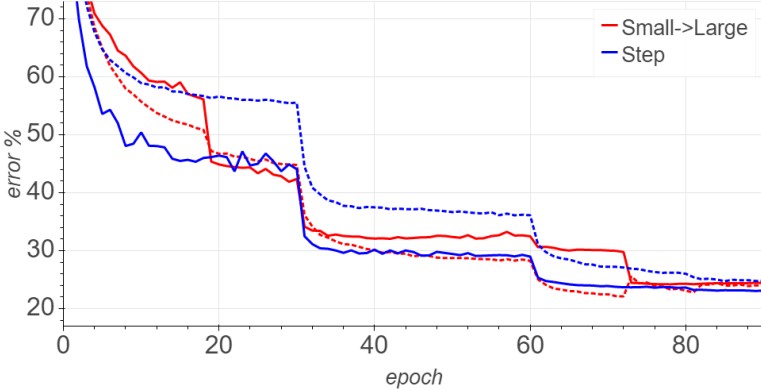

Figure 9: Test accuracy vs epoch (ResNet50 on ImageNet, $B^+$ with $s^{(144)}$) for progressive resizing (small to large) compared to per-step sampling.

$$S^{(208)}: \quad S = \begin{cases} 320, & \text{w.p} \quad p = 0.1 \\ 288, & \text{w.p} \quad p = 0.1 \\ 256, & \text{w.p} \quad p = 0.1 \\ 224, & \text{w.p} \quad p = 0.2 \\ 192, & \text{w.p} \quad p = 0.2 \\ 160, & \text{w.p} \quad p = 0.1 \\ 128, & \text{w.p} \quad p = 0.1 \\ 96, & \text{w.p} \quad p = 0.1 \end{cases}$$

