# OpenReview forum: "MixSize: Training Convnets With Mixed Image Sizes for Improved Accuracy, Speed and Scale Resiliency"
_ICLR.cc/2021/Conference — Reject_

### Official Review · AnonReviewer1 · 2020-10-24
**A mixture of multiple techniques without ablation**

**Rating:** 5
**Confidence:** 3

**Review:**

The paper proposes the use of mixed image sizes during training. They argue empirically that such an approach improves generalization for both fixed image size (e.g. 224 in ImageNet) as well as for variable image size. The proposed training algorithm maintains the same computational budget at each step by either changing the batch size or by using more/less augmentation. They show that adjusting the batch size leads to a faster training but using augmentation leads to a better test accuracy (hence a tradeoff).

However, in order for their proposed method to work, the authors also propose modifications to standard training procedures (i.e. smoothing the gradient, adjusting the batchnorm layers) but without carrying out an ablation study that shows the impact with/without each of these steps. My particular concern is in the use of "gradient smoothing". If I understand it correctly, this is not very different from using momentum, which is known to reduce the variance and improve generalization. However, the authors use gradient smoothing only in their proposed method and do not use it in the baseline method (why not?). It is possible that the reported improvements (e.g. for fixed image size) come solely from this step.

The other concern is when sampling the image size per step. The authors  propose distributions that seem odd in their experiments (e.g. why is p=0.6 for size 128 in ImageNet which is much larger than others, and why not uniform in CIFAR10). It is important to know if the results are sensitive to the choice of the distribution, to make sure that the benefit is not due to random chance. Also, if this distribution needs to be fine-tuned, then the discussion about improving the training time would be meaningless.

The last issue is the robustness to different image sizes. Figure 3(a) shows  that if the average image size during training is small, the network will perform better for small images but not for large images. Conversely, if the average image size during training is large, it will perform better for large images, but not for small images. If the concern here is around using mixed-image sizes at inference time, then the red curve in Figure 3(a) shows that a fixed image size is reasonably robust. If one knows in advance that the average image size would be smaller than 224, one can train with a fixed image size that is smaller.

One minor last remark (feel free to ignore) regarding the motivation: the authors study the correlation between the full gradients for the same image with different sizes, on the one hand, and for different images with the same size, on the other hand. They conclude that the first case (different sizes) shows a stronger correlation, which is true according to Table 2, but this statement omits the fact that most correlations were low anyway. For example, for partially trained network, it is 0.08 vs. 0.02. I do not think that one can use such figures to conclude that "smaller image gradients can be used as an estimate to the full image gradients".

The improvement in test accuracy is very promising but I believe, some ablation is needed to identify exactly where this improvement comes from and whether it can be obtained using simpler approach (e.g. smoothing the gradient alone or using augmentation alone, etc).

---

> ### Author Response · Authors · 2020-11-23
> **answer**
>
> We thank the reviewer for the detailed and helpful feedback.
> Regarding gradient smoothing (GS) vs momentum -- the key aspect of this method is to reduce the amount of noise in gradient norm, as it varies when computed using different batch-sizes in each step. One way to think of it is as a momentum factor over gradient norm, without impacting gradient direction. GS is not required when batch size is fixed throughout training as in baseline or D+ regime. We will make the distinction clearer between GS and momentum.
> Sampling image size -- the choice for sampling probability was guided by the need to reduce the average image size, to allow increased batch/duplicate factor. To select the most promising regimes, we simply cross-validated from a pool of 4-5 options. We will add more info regarding this process.
> Regarding robustness to different image sizes --- this is a good point and another value of our work, as it allows us to design a training regime targeting inference performance of a range of possible sizes. So while fixed image size at training may allow good results over a narrow range around that size, we can improve it by offering better coverage and ensure good performance even when inference image size varies (same model for different sizes).
> We thank the reviewer for his remark on correlation measurement and will add this reservation.

---

> > ### Comment · AnonReviewer1 · 2020-11-24
> > **Gradient Smoothing**
> >
> > Thank you for the response.
> >
> > To clarify my concerns, regarding gradient smoothing, the primary concern was not about how similar/dissimilar it was to momentum. The main concern was that it is a noise reduction method that was used *only* for the proposed algorithm, but not the others, even though it could be used for the other algorithms as well. It is possible that the reported improvements (e.g. for fixed image size) came solely from GS.
> >
> > Also, the concern about robustness is not at inference time. My concern was during training. Will the performance significantly change if you change the sampling distribution during training?

---

> > > ### Author Response · Authors · 2020-11-24
> > > **Gradient Smoothing**
> > >
> > > Thanks for the clarification.
> > > We tested GS on the other methods (baseline and D+), and it did not improve accuracy. We therefore stated that it was beneficial only for B+ regime (where it improved over B+ without gradient smoothing and over baseline).
> > >
> > > Regarding robustness at training --- in all sampling regimes we tested performance was better than a fixed size training with the same average size. The regimes shown in the paper were those that had the best robustness to size variation at inference. We will add all these intermediate results to the Appendix.

---

### Official Review · AnonReviewer4 · 2020-10-27
**Not a bad paper but lacking of significant contributions**

**Rating:** 5
**Confidence:** 5

**Review:**

This paper proposes to increase training costs to compensate for the reduced costs from multi-scale CNN training by either increasing batch size (and therefore lowering the number of iterations per epoch) or increasing the number of augmented versions (duplicates) of the same samples within a batch. The former allows for smaller total training costs than conventional single-scale training, while the latter maintains the total training costs but improves the final performance. Several training improvement methods are introduced to improve the multi-scale training.

Paper's strengths
- The paper is quite well-written.
- Code and models are provided for reproducibility.
- Gradient smoothing is a nice way to mitigate the variability of gradient statistics resulted from different input sizes. As far as I know, this is quite novel, particularly in the context of multi-scale training.

Paper's weaknesses
- Multi-scale training is a common practice in many computer vision tasks especially in object detection** (less common in image classification). This paper also does multi-scale training but only introduces some minor improvements that are neither breakthroughs nor that they provide any interesting insight.
> - Bag of Freebies for Training Object Detection Neural Networks. arXiv.
> - YOLO9000: Better, Faster, Stronger. CVPR 2017.
> - MMDetection: Open MMLab Detection Toolbox and Benchmark. arXiv.

- For "step-wise size sampling", it seems like that conclusion to use this variant of sampling is heuristically chosen and totally ignores the existing practice in other computer vision tasks. One of the straightforward ways to do multi-scale training in object detection is to select different input sizes even for the images within the same batch (by padding zeros for the smaller images). Alternatively, one could sample different input sizes for different GPU batches (all images within a GPU share the same size) while doing multiple-GPU training.

- The three training improvements (step-wise size sampling, gradient smoothing and batch-norm calibration) are what separate this paper from prior work but they are not extensively evaluated. Some of them are briefly evaluated in the appendix and some of their effects are just briefly mentioned in the method section. They ought to appear in the experimental section of the main paper. Gradient smoothing seems like a nice idea but it is unclear how important it is given that there is only one figure (Fig.7) showing its impact on the performance.

- This paper strives to increase the number of batch samples given a fixed budget of computational and time resources for per iteration step. I wonder why this should be limited to the cost of an iteration step but not the entire training cost from all epochs/iterations. It makes more sense to measure the cost for the entire training process which accurately tells how much is spent to train the model until convergence.

- In Sec. 5.1, the size ratios for different datasets are carefully chosen based on cross-validation. This makes it hard to directly apply MixSize to other datasets or settings without going through this step. It also adds additional computations which defeat the purpose of increasing batch size to maintain the same training cost as conventional single-scale training.

- Using separate BatchNorm statistics for multi-scale inputs/features has been explored in the following papers (published at least few months before ICLR deadline). They should be cited and compared against MixSize:
> - Learning to Learn Parameterized Classification Networks for Scalable Input Images. ECCV 2020.
> - Stochastic Downsampling for Cost-Adjustable Inference and Improved Regularization in Convolutional Networks. CVPR 2018.

Overall, this paper shows good performance and has some good ideas (e.g., gradient smoothing) for improving multi-scale training. But it fails to give more emphasis to or do a deeper dive into the potentially novel aspects of the work. The current performance improvements may come from doing just trivial multi-scale training which was already widely-explored in prior work.

---

> ### Author Response · Authors · 2020-11-23
> **answer**
>
> We thank the reviewer for the detailed and helpful feedback.
> As the reviewer mentioned (and also appears in our background section 2.1), multiple image sizes are used in detection and segmentation models. However, our work deals with modifying image size as part of the training regime for classification models ( which can later be used as backbone for detection). "Padding" technique suggested by the reviewer is irrelevant for our use case, as it will not allow us to save the computational cost of the model (zero values will be computed).
> As suggested, we will add additional info on our training improvements.
> As mentioned in section 5.2, wall-clock times scales similarly to steps, as we balanced compute (through batch size) for all configuration. Furthermore, we kept the number of epochs fixed for all regimes, where D+ regime keeps the same number of steps (without enlarging batch-size).
> Missing previous work - we thank the reviewer and will add the missing recent studies. We note, however, that all mentioned works are suggesting a modified model structure. In this work, we suggested a modification to the training technique, such that existing models can benefit from the use of multiple image scales. We've added a comparison with previous works to Appendix.

---

### Official Review · AnonReviewer2 · 2020-10-30
**A nice trick, but not enough novelty**

**Rating:** 5
**Confidence:** 4

**Review:**

== Summary ==

The paper proposes to use different image resolutions during the training of a deep neural network image classifier, and varying the batch size or number of data augmented versions of the images, keeping the computational cost per step roughly constant. The authors apply this approach to several architectures and three datasets, and show that they can achieve or improve the same accuracy as the baselines but much faster; or achieve better results with the same computational budget.

== Pros ==

+ The authors conducted their experiments using three different datasets (Cifar10, Cifar100 and ImageNet), and six different architectures (ResNet44, ResNet50, WideResNet, DenseNet and EfficientNet).

+ The proposed approach outperforms the baselines consistently across the 8 pairs of (dataset, architecture) that they have studied. In addition, MixSize can be easily implemented (the authors also provide a PyTorch implementation).

+ The authors investigate different "tricks" to apply during training when using MixSize to stabilize training or achieving better results. For instance, they compared randomly sampling the size from a distribution (as they propose) versus increasing the image resolution as training progresses, and showed that randomly sampling yields slightly better results.

+ Figure 3a seems to suggest that MixSize yields more robust classifiers under a wide range of image sizes. The area under the "Mixed S=144" curve seems to be larger than the area under the "Fixed S=224". However, further experimentation is needed to confirm this, since the area of the "Mixed S=208" seems closer to "Fixed S=224", and in any case the maximum image size was capped around 415.

== Cons ==

- On of the claimed contributions is: "Faster training vs. high accuracy. We show that reducing the average image size at train- ing leads to a trade-off between the time required to train the model and its final accuracy.". However, I would not consider this a novel contribution, since the trade-off between speed and accuracy is well-known. In fact, the authors cite Huang et al. (2018) and Szegedy et al. (2016) which already showed this. EfficientNet is another well-known architecture that takes advantage of this fact.

- In the intro, the authors claim "Touvron et al. (2019) demonstrated that networks trained on specific image resolutions perform poorly on other image sizes at evaluation time, as confirmed in Figure 1". This is inaccurate, since Touvron et al. (2019) actually show that slightly increasing the test resolution improves accuracy, due to the discrepancy in object sizes introduced by data augmentation (cropping). In fact, Figure 1 shows the same effect (model trained with 224 res, achieves best results with 284 eval image size). The statement in the introduction is again contradicted at the end of the first paragraph in Section 2.1.

- The authors do not report any statistical significance metric. Some datasets have very close results, so it's hard to tell whether the improvements are (statistically) significant or not.

- Poor captions in figures and tables. For instance, the difference between solid lines and dashed lines is only explained at the very last figure in the appending (Figure 7). Also, the caption of Figure 3 reads "Test accuracy on validation set" which is ambiguous: Is it a typo, or is it that the authors report the results on the 50k validation set of ImageNet (and use some smaller subset of the training set as validation)?

== Reasons for score ==

Although the proposed approach is simple and consistently improves the baseline results, I'm not convinced that the originality and significance of the work is enough for it to be accepted.

Regarding originality, there is a plethora of works exploring the trade-offs between image size and accuracy. The most similar works are Howard (2018) and Karras et al. (2017), which increase the image size through training. It's not clear that random sampling offers a much better result, judging from Figure 8 in Appending E, if one compares the accuracy of "Small->Large" strategy at 125k steps (possibly before the last increase in size).

Regarding significance, if one restricts the analysis to the best architectures in each dataset, the increase in accuracy does not seem to be very large. Cifar10: 98.16% -> 98.32% (AmoebaNet), ImageNet: 76.32% -> 76.53% (EfficientNet-B0). Cifar100 shows a larger improvement, but the authors did not use AmoebaNet (which worked best in Cifar10) for some unknown reason. The fact that no statistical significance metrics are reported, does not help to discern whether the improvements are meaningful or not.

---

> ### Author Response · Authors · 2020-11-23
> **answer**
>
> We thank the reviewer for the detailed and helpful feedback.
> On "trade-off between speed and accuracy": We note that all mentioned works are suggesting a modified model structure. In this work, we suggested a modification to the training technique, such that existing models can benefit from the use of multiple image scales. Thus we were first to show trade-off in training regime for, rather than model size.
> We thank the reviewer for his comment regarding improved resolution with increased size as cited from Touvron et al. (2019). What we meant was that beyond a small increase over trained size (for which accuracy may improve), we see clear degradation of the model, which cannot be (currently) fixed without fine-tuning as Touvron et al. suggested. We will make this more clear.
>
> We will make captions more clear. As for Figure 3, we measured accuracy on ImageNet's validation set, training was done on all training set (no additional validation split here)
> We will add statistical significance scores to the results.

---

### Official Review · AnonReviewer3 · 2020-11-02
**An interesting paper, but needs improvements**

**Rating:** 5
**Confidence:** 5

**Review:**

This paper presents a mixed-size CNN training scheme, using several different input image sizes for one single model training. The authors assume the training budget, represented as S_i^2*B_i*D_i (i.e., spatial sample size, the number of batched distinct samples and the duplicates for each distinct sample), to be a fixed constant during training step i.  Under such an assumption, two mixed-size training scenarios are considered, one for training acceleration and the other for improved model generalization ability. The authors additionally use step-wise image size sampling, gradient smoothing, and per-size BN calibration to enhance the model performance under the above two mixed-size training scenarios. Experimental validation is performed on CIFAR and ImageNet datasets using diverse CNN structures.

Mixed-size training is a critical problem and the methods proposed in this paper are interesting. My main concerns to this paper are as follows.

--- Critical related works and comparison are missing.

Mixed-size training of CNNs for image classification are not new. Here are some recent works, however they are missed by the authors.

“Resolution Adaptive Networks for Efficient Inference”, in CVPR 2020

 “Resolution Switchable Networks for Runtime Efficient Image Recognition”, in ECCV 2020.

“MutualNet: Adaptive ConvNet via Mutual Learning from Network Width and Resolution”, in ECCV 2020.

Besides, as described by the authors, NeurIPS 2019 paper “Fixing the train-test resolution discrepancy” also considers how to enhance the performance of CNN models when applying them with different input image sizes. But a performance comparison is missing.

To show the advantages of this paper, thorough discussion and performance comparison with the above works are necessary. Taking ResNet-50 trained on ImageNet as an instance, I notice that the proposed method shows obviously worse accuracy compared to some of these works.

---Another critical baseline is also missing.

In page 8, “We note that for original fixed-size regimes this calibration procedure resulted, with degraded results and so we report accuracy without calibration for these models”. To my understanding, this is somewhat weird. Why BN calibration does not work on the other image sizes when the model is trained with a fixed image size? It is not clear. Furthermore, in NeurIPS 2019 paper “Fixing the train-test resolution discrepancy”, this line of methods work pretty well. Such a BN calibration should server as another baseline for more fair comparison.

---Regarding B+ design

How about the wall-clock training cost (in hours) instead of the number of iterations/epochs?
How about the performance of applying “scale the learning rate linearly” to train the baseline model?

---How about model transfer ability?

Only image classification tasks are considered in the experiments. How about the performance of the trained backbone models, when transferring them in the downstream tasks, such as object detection and semantic segmentation? Can performance gain be transferred?

---Others

Is there any principled way regarding the size sampling strategy? The current strategy is based on manual setting, which limits its use in real applications.

I suggest the authors to also provide precise accuracy numbers, etc. regarding some figures (e.g., figure 1, figure 4) shown in the paper.

Generally, I am on the fence, to this paper. I encourage the authors to address the questions I raised above.

---

> ### Author Response · Authors · 2020-11-23
> **answer**
>
> We thank the reviewer for the detailed and helpful feedback.
> Answering raised concerns:
> 1) Missing previous work - we thank the reviewer and will add the missing recent studies. We note, however, that all mentioned works are suggesting modified model structure (thus similar to works we cited in the introduction -Takahashi et al., 2017; Xu et al., 2014; Zhang et al., 2019). In this work, we suggested a modification to the training technique, such that existing models can benefit from the use of multiple image scales. We've added a comparison with previous works to Appendix.
>
> 2) BN calibration does not work for fixed scale trained networks - we do not know the reason for this property (calibration on these models simply yielded worse results and so were not used on baseline).  We speculate that changing image scale post-training simply requires finetuning to the network beyond first and second statistical moments (which BN calibration allows) and so light parameter training is required (as done by Touvron et al.)
>
> 3) As mentioned in section 5.2, wall-clock times scales similarly to steps, as we balanced compute (through batch size) for all configuration. We will add the exact number in hours.
> 4) Measuring our models on subsequent detection tasks did not show significant improvement (performed similarly to baseline), we will add this info to our work.
> 5) We currently do not have any principled way to determine sampling size, but found cross-validation to be effective to select best-performing regimes.

---

### Decision · Program_Chairs · 2021-01-07
**Final Decision**

**Decision:**

Reject

**Comment:**

This work proposes to train networks with mixed image sizes to allow for faster inference and also for robustness. The reviewers found the paper was well-written and appreciated that the code was available for reproducibility. However, the paper does not sufficiently compare to related methods. The authors should resubmit once the comparisons suggested by the reviewers have been added to the paper.